# Peptide-Mediated Nanocarriers for Targeted Drug Delivery: Developments and Strategies

**DOI:** 10.3390/pharmaceutics16020240

**Published:** 2024-02-06

**Authors:** Yubo Wang, Lu Zhang, Chen Liu, Yiming Luo, Dengyue Chen

**Affiliations:** 1Medical College, Guangxi University, Da-Xue-Dong Road No. 100, Nanning 530004, China; ybwang9315@163.com; 2School of Life Sciences, Xiamen University, Xiamen 361005, China; 19550996705@163.com; 3Fujian Provincial Key Laboratory of Innovative Drug Target Research, School of Pharmaceutical Sciences, Xiamen University, Xiamen 361005, China; 15063035260@163.com; 4Department of Hematology, The First Affiliated Hospital of Xiamen University and Institute of Hematology, School of Medicine, Xiamen University, 55 Zhenhai Road, Xiamen 361003, China; 5The School of Clinical Medicine, Fujian Medical University, Fuzhou 351002, China

**Keywords:** peptides, self-assembly, peptide–drug conjugates, drug delivery, peptide modification

## Abstract

Effective drug delivery is essential for cancer treatment. Drug delivery systems, which can be tailored to targeted transport and integrated tumor therapy, are vital in improving the efficiency of cancer treatment. Peptides play a significant role in various biological and physiological functions and offer high design flexibility, excellent biocompatibility, adjustable morphology, and biodegradability, making them promising candidates for drug delivery. This paper reviews peptide-mediated drug delivery systems, focusing on self-assembled peptides and peptide–drug conjugates. It discusses the mechanisms and structural control of self-assembled peptides, the varieties and roles of peptide–drug conjugates, and strategies to augment peptide stability. The review concludes by addressing challenges and future directions.

## 1. Introduction

Cancer encompasses a variety of diseases characterized by uncontrolled, abnormal cell growth, driven by mutations that alter growth signal responses in response to growth signals, defects in the cell death program, and an increased ability to form new blood vessels and invade tissues. The diverse nature of cancer sites and mutations presents challenges to selectively eliminating the cancer cells and preserving normal cells. In recent years, advances in nanomedical technology have opened new avenues for the targeted delivery of antitumor drugs. Nanomedicine drug delivery systems offer significant potential to improve the pharmacokinetic behavior of insoluble drugs, reduce the off-target effect of anticancer drugs, increase drug stability, and achieve precise drug delivery and controlled release. Nanomedicine drug delivery systems have shown great potential in drug-targeted tumor delivery [1].

An important focus of nanomedicine research involves targeting drugs to specific organs, target cells, or even organelles to achieve precise drug delivery, improve the drug effect, and reduce the dose and toxic side effects. Commonly used targeting strategies include passive targeting of tumor tissues by EPR effect and targeted delivery of drugs by using highly expressed or specifically expressed receptors at the target site. In the research of targeted therapy, offering advantages over traditional large, complex, and immunogenic antibodies with large molecular weight, complex structure, and immunogenicity, targeted peptide small molecules have received more and more extensive research attention. Peptides, as emerging nano-drug components, can self-assemble to form fine nanostructures, are amenable to targeted modifications and responsive drug release, and have the advantages of high drug-carrying capacity, small molecular weight, low immunogenicity, and low preparation cost, which provide a new research direction for the construction of intelligent nano-delivery systems (Figure 1).

Peptide-mediated tumor-targeted delivery systems offer distinct advantages compared to both free drug and passive tumor-targeting delivery systems. They actively recognize tumor-specific receptors, leading to better tumor targeting, enhanced retention, and increased uptake by tumor cells. Additionally, these delivery systems demonstrate a superior ability to overcome tumor multidrug resistance. Peptide-mediated systems present further advantages over monoclonal antibody-mediated systems, including simple preparation method, low immunogenicity, suitable stability, and easy delivery [2]. However, peptide-mediated nano-delivery systems still face challenges. They may encounter increased capture by the reticuloendothelial system, resulting in broader distribution in the liver and spleen, potentially compromising their function. The variability of tumor tissue receptors and the presence of peptide degradation enzymes in the body can affect the efficiency and targeting performance of these systems. Although peptide modification holds promise in improving targeting efficiency, its absolute distribution in tumor sites is still low [3]. Further research is needed to enhance the targeting efficiency of peptide-mediated tumor-targeted nano-delivery systems.

This review firstly explores the use of self-assembled peptides as building blocks for nano-delivery systems and their assembly driving force, then reviews the progress in intelligent nano-drug delivery systems constructed by peptides with various functions (targeting, penetrating, responsive, etc.). Finally, the discussion includes strategies to enhance peptide stability and provides an outlook on peptide-mediated drug delivery systems.

## 2. Self-Assembled Peptides for Drug Delivery Systems

Molecular self-assembly is the process by which molecular entities autonomously and reversibly form a stable arrangement of a specific shape in thermodynamic equilibrium and without external intervention [4], which is a widespread phenomenon in nature. Self-assembly plays an important role in fields ranging from protein folding to surface chemistry and nanotechnology (Figure 2).

Peptides are ideal for constructing nanodevices because of their simple structural composition, versatility, and modifiability. Amino acid monomers consist of amino and carboxyl functional groups as well as unique side chains. The differences in amino acid side chains allow them to be classified into different categories based on their chemical composition, such as hydrophobic, hydrophilic, and fatty chains. Peptide molecules are a class of compounds with a structure intermediate between amino acids and proteins, which are both structural fragments that constitute proteins and active fragments in which proteins function [5,6,7]. Peptide molecules, with different amino acid sequences, formed nano- and microscale structures with certain spatial conformations through self-assembly to obtain a series of biological functions.

As a novel field developed in recent decades, peptide self-assembly has a very broad research space [8] and offers significant potential for developing self-assembled biomaterials. Unlike complex biomolecules like proteins, peptides’ shorter sequences facilitate synthesis and modification; additional chemical groups can be conveniently inserted in the process of research to design novel materials with higher flexibility and complexity in terms of structure and function [9,10].

### 2.1. Drivers of Peptide Self-Assembly

Amino acids provide the primary structure and site for chemical modification when designing peptide nanomaterials. Amino acid side chains have different charges, hydrophobicity, sizes, and polarity (Table 1). The number, type, and sequence of amino acids can be manipulated to design unique self-assembled peptide nanostructures with specific secondary structures and physicochemical properties [11,12,13,14].

Peptides can spontaneously aggregate in aqueous solution by utilizing differences in the hydrophilicity of amino acid side chains. Among them, peptide amphiphilic molecules (PAs) have introduced hydrophobic chains based on peptides formed by condensation of amino acids self-assemble into novel nanostructures. Their assembly is governed by amphiphilicity, with a common structure comprising a secondary structure-forming peptide fragment and a hydrophobic tail. PAs spontaneously form high aspect ratio nanostructures and generate structurally well-defined self-assembled nanostructures through unique intramolecular and intermolecular interactions under certain pH, temperature, and ionic strength conditions.

#### 2.1.1. Hydrogen Bonds

The self-assembly of peptide amphiphiles (PAs) is driven by hydrogen bonding, resulting from electron acceptor hydrogen atoms (N-H) and electronegative donor oxygen atoms (C=O) on the backbone of the peptide molecule, resulting in hydrogen bonding [15]. Hydrogen bonding influences peptide arrangement and facilitates axial extension to form an ordered secondary structure. It is categorized into intermolecular and intramolecular bonding: intermolecular hydrogen bonding and intramolecular hydrogen bonding, and each leads to different structures. Intramolecular bonding within long peptide chains, the formation of α-helix is facilitated by intramolecular hydrogen bonding formed by the interaction of internal amino acid side groups. Intermolecular bonding facilitates the formation of β-sheet structures [16]. Paramonov et al. [17] prepared a dodecapeptide with a C_16_ end and determined the role of hydrogen bonding at different positions for the assembly of the peptide by disrupting amino acid hydrogen bonding in the hydrophobic core. They demonstrated that hydrogen bonds closer to the hydrophobic core are crucial for nanostructure stability; the morphology of the nano-assemblies changed from fibrous to globular when the hydrogen bonds at this position were broken. Meanwhile, breaking bonds near the hydrophilic end has less impact on the morphology of the peptide assemblies.

#### 2.1.2. Hydrophobic Interactions

Hydrophobic interactions are key drivers for PA self-assembly. It refers to the entropic effect, where nonpolar surfaces with polar solvents bind to minimize surface exposure, thus maximizing entropy [18]. PAs aggregate through hydrophobic interactions in the terminal hydrophobic region, which, in turn, strengthens the formation of hydrogen bonding, in which the peptide molecule exposes the hydrophilic portion of the interface to form a specific nanostructure. The strength of the hydrophobic interaction plays a very important role in the stabilization of the peptide secondary structure. Therefore, the aggregation morphology of peptide nanostructures can be changed by adjusting the strength of the hydrophobic portion. Löwik et al. [19] investigated the effect of the length of the terminal alkyl chain of an octapeptide on the molecular assembly; they found that an increase in the terminal hydrophobic chain favored the formation of the nano-assemblies under the same conditions and that the longer the alkyl chain was, the more stable was the secondary structure of the peptide assembly.

#### 2.1.3. Electrostatic Interactions

Electrostatic forces are long-range, monounsaturated, and non-directional and typically act between charged amino acid residues [20]. Electrostatic forces in peptide assembly systems are divided into two categories: (1) peptides with the same charge repel, preventing aggregation; (2) oppositely charged peptides attract, forming ordered aggregates. The aggregation morphology can be effectively regulated by modulating the number, type, and distribution of charges of the peptides. Papapostolou et al. [21] designed two complementary peptides and promoted self-assembly through electrostatic interactions using zipper-like growth, which led to the formation of hexagonally arranged nanofibers.

#### 2.1.4. π-π Stacking Interactions

π-π stacking interactions, occurring between aromatic rings and are often introduced by aromatic amino acids, mostly associated with the presence of aromatic amino acids such as phenylalanine, tyrosine, and tryptophan. These interactions are common in amyloid peptide sequences; the introduction of these amino acids not only enhances the hydrophobic interaction between the benzene rings but also provides π-π stacking interaction [22], which greatly facilitates peptide self-assembly. The simplest self-assembly system involves a dipeptide formed by two phenylalanine, and Carny et al. [23] used the π-π stacking interaction between phenylalanine side chains to form stable nanotubes and hydrogels in aqueous solutions.

### 2.2. Secondary Structure in Peptide Self-Assembly

The primary structure of a peptide molecule refers to its amino acid sequence, and when this main chain coils or folds in specific ways, a specific secondary structure can be formed, a characteristic feature of peptides distinguishing them from other organic polymers [24]. These secondary structures enable further development into advanced assemblies. The secondary structures of peptide self-assembly generally include more stable α-helix, β-sheet, β-turn, and unstable random coil structures.

#### 2.2.1. α-Helix Structure

The α-helix is a prevalent secondary structure in peptides. Characterized by a regular helical arrangement of the main chain around a central axis, with 3.6 amino acid residues per turn, i.e., C=O in the amide bond will form a hydrogen bond with the fourth N-H [25,26]. Based on the chiral nature of peptide formation from natural amino acids, the α-helix structure appears as a right-handed helix, which is usually stabilized by intramolecular hydrogen bonding. α-Helices are influenced by amino acid sequences and the balance of hydrophilic and hydrophobic residues.

The α-helix structure is commonly found in signal peptides, membrane proteins, and proteins with membrane recognition functions and is also typical of many membrane-penetrating peptides. Yao et al. [27] correlated the cell-penetration ability of peptides with the degree of α-helix structure and hydrophobicity and showed that peptides were attractive for in vivo delivery because of their low cytotoxicity, low hemolysis, and suitable substance transport properties. The low hemolytic and better substance transport properties of peptide molecules have become attractive tools for in vivo delivery. Zhu et al. [28] observed that substituting the 19th amino acid in a 26-residue α-helix peptide with glycine reduced the helix’s strength and, consequently, its cytotoxicity.

#### 2.2.2. β-Sheet Structure

The β-sheet consists of extended peptide chains, which are generally formed by the joint participation of two or more peptide chains. Regular hydrogen bonds can be formed between the amino and carbonyl groups on the main chains of adjacent peptide chains, which, in turn, form a naturally twisted lamellar structure, which can be arranged in either a parallel or antiparallel arrangement. When the peptide chains are stacked to form a parallel β-sheet structure, there is an angular deviation in the hydrogen bonds between the molecular chains; when the peptide chains are stacked to form an antiparallel β-sheet structure, the hydrogen bonds between the molecular chains are parallel to each other. Since the intermolecular hydrogen bonds formed when the peptide chains are parallel are unstable with high energy, most of the β-sheet structures are inclined to be antiparallel [29,30]. β-sheet structures have obvious negative characteristic peaks at 218 nm in the circular dichroism spectrum.

A further understanding of the formation of nano-assemblies from β-sheet structures has led to the interest of several researchers in nano-assemblies containing the same secondary structure but exhibiting different morphologies. Singh et al. [31] observed a morphological transformation in a pentapeptide from spherical to helical nanofibers, showing that peptide assembly resembles reactive polymerization with controllable process and morphology. During the morphological transformation, the authors found that the assembly process of the peptide molecules was similar to reactive polymerization and that the polymerization process and morphology could be controlled by the continuous addition of monomers.

#### 2.2.3. β-Turn Structure

In β-turns, often connecting α-helices and β-sheets in proteins, the C=O of the first amino acid at the β-turn forms a hydrogen bond with the N-H of the fourth amino acid to stabilize the turn [32,33]. β-turn structures consist of four consecutive amino acid residues nearby, which usually contain proline and glycine as well as polar amino acids. The absence of a glycine side chain reduces spatial resistance, and the unique structure of the proline forces the formation of the β-turn. β-turn structures have a distinct positive characteristic peak at 206 nm in the circular dichroism spectrum. Robinson [34] created cyclic peptides with β-turn structures using a hairpin template. This template, devoid of the disulfide bonds typically associated with the β-turn, established a stable β-turn structure through hydrogen bonding between the amino acids. The hydrogen bonds among amino acids contribute to the stability of the formed β-turn structure.

#### 2.2.4. Random Coil

Random coils represent an irregular, disordered state of peptide chains, considered high in conformational energy and instability [35]. Despite their disordered nature, random coils can form specific self-assemblies. This structure has a pronounced negative characteristic peak at 200 nm in the circular dichroism spectrum. Guler and Stupp [36] designed peptides with varying structures and observed that the removal of hydrophobic chains and incorporation of proline reduced β-sheet formation, and the authors synthesized peptide 3, which does not have a hydrophobic chain by incorporating proline into the end of the peptide. The authors also synthesized peptide 3 without a hydrophobic chain and doped proline into the peptide sequence to form peptide 2 and peptide 4. The experimental results showed that the removal of the hydrophobic chain and the introduction of proline reduced the ability of the peptide molecules to form β-sheet structures, leading to random coil structures. Among the peptides studied, only the β-sheet-forming peptide showed high catalytic activity.

### 2.3. Classification of Peptide Self-Assembly

Peptide self-assembly, as previously discussed, is the result of the interaction of intermolecular forces within them. When the forces reach a stable state of relative equilibrium, a regular and orderly nanostructure is formed. This equilibrium is affected by changes in the external environment. Based on the influence of external factors, peptide self-assembly can be divided into two types: spontaneous self-assembly and triggered self-assembly.

#### 2.3.1. Spontaneous Self-Assembly

This type occurs when peptides form self-assemblies upon dissolution in an aqueous solution without external stimuli. Vauthey et al. pioneered the study of surfactant-like peptides, designing a series of self-assembling short peptides A_6_D, V_6_D, V_6_D_2_, L_6_D_2_, etc. [37], which have multiple hydrophobic tails composed of multiple hydrophobic amino acids (alanine A, valine V, leucine L, etc.) and heads composed of hydrophilic amino acids (aspartic acid D) [38]. Resembling the structure of a phospholipid bilayer, in an aqueous solution, the hydrophilic head of the two peptides is distributed outward, and the hydrophobic tail is distributed inward, which interact spontaneously to form a closed circle, after which the circles are stacked sequentially; finally, nanotubes and vesicles with diameters of 30–50 nm are formed. Cavalli et al. reported an amphiphilic lipopeptide, ALPs [39], whose hydrophilic-terminal oligopeptides are composed of multiple alternating leucine (L) and glutamic acid (E), with succinyl as an intermediate connecting part to the hydrophobic phospholipid tail, which can spontaneously form monomolecular lamellae with β-sheet at the air–water interface.

#### 2.3.2. Triggered Self-Assembly

Triggered self-assembly occurs in response to when the external environment changes. The force between peptide molecules changes accordingly, and self-assembly occurs by changing its structure or state to adapt to environmental changes. This dependence on the environment provides peptide self-assembly materials with suitable controllability. They become multifunctional materials capable of making rapid and precise responses to changes in the external environment, particularly in biomedical applications in the field of biomedical materials.

##### pH-Triggered Self-Assembly

Changes in the environmental pH (both external and physiological pH) cause protonation and deprotonation of basic and acidic amino acids, affecting the electrostatic interactions between the peptide molecules and the self-assembly state of the peptides. Altman et al. designed a series of peptides that can complete the reversible transformation of α-helix and β-sheet secondary structures at different pH [40]. Chen et al. designed a pH-controlled amphiphilic peptide self-assembly system [41]. Nanofiber structures were generated by designing electrically complementary peptide sequences composed of amino acids with opposite charges to regulate inter- and intra-fiber interactions. In the pH-induced alternation of positive and negative charges on the peptide surfaces, the complementary-attractive arrangement of charged residues leads to staggered nanofibers and “bundled” structures, while the complementary-repulsive arrangement of charged residues leads to well-dispersed nanofibers.

##### Temperature-Triggered Self-Assembly

Peptide self-assembly is also sensitive to temperature changes. Elevated temperatures disrupt hydrogen bonding, leading to the unfolding of its secondary structure, altering the self-assembly mechanism, and affecting the functional rate of self-assembly. Rughan et al. found that the peptide MLD was induced to self-assemble when the temperature was elevated, and the secondary structure of the peptide was transformed from an irregularly curled to a β-sheet, which led to the formation of a crosslinked fibrous network structure and the formation of a hydrogel, which was able to rapidly recover to its original strength after being damaged by external forces [42]. Altma et al. showed that temperature influences the transition between the α-helix and β-sheet of the peptide secondary structure [40].

##### Light-Triggered Self-Assembly

In nature, light plays the roles of biocatalyst, information, and energy transport. The incorporation of photosensitive groups into peptides can have a great impact on the self-assembly behavior of peptides. Haines et al. [43] designed a photoexcited hydrogel system that uses light to trigger the self-assembly of water-soluble peptides into hydrogel materials. The light-triggered small peptides undergo folding to form amphiphilic hairpin structures that can be efficiently and rapidly self-assembled into hydrogels with mechanical strength. Chen et al. [44] connected two α-helix peptides with azobenzene, which undergoes reversible transitions by alternating visible and UV light irradiation, affecting the peptide’s molecular structure and self-assembly.

##### Receptor–Ligand Binding-Triggered Self-Assembly

Enzyme-triggered peptide self-assembly is chemically, regionally, and enantioselectivity advantageous [45,46]. Matrix metalloproteinases (MMPs) belong to a large family of secreted proteases that play important roles in normal physiological processes. MMPs overexpressed in many malignant tumors and regulated peptides in tumor treatment. Straley and Heilshorn prepared elastin-like polymers that contain peptide regions that are readily cleaved by cell-secreted proteases [47]. Todd et al. investigated the RGD sequence’s hydrolysis by enzymes [48]. Yang et al. [49] synthesized a hydrogel factor, naphthalene acetic acid-FFGEY, capable of regulating its assembly through phosphokinase/phosphodiesterase, affecting the peptide’s phosphorylation and dephosphorylation. The addition of phosphokinase phosphorylates the peptide and destroys the self-assembled structure, and the addition of phosphodiesterase for dephosphorylation restores the peptide to its ordered nanostructure.

### 2.4. Peptide Amphiphilic Molecules Self-Assemble into Structures

Peptide amphiphilic molecules form assemblies by self-assembly, involving four components: (1) hydrophobic chain tails that provide hydrophobicity; (2) peptide fragments that can form secondary structures; (3) hydrophilic fragments enhancing molecular solubility; and (4) the introduction of functional groups [50]. Despite their structural simplicity, linear peptides can form diverse morphologies such as nanofibers [51], nanosheets [52], nanotubes [53], vesicles [54,55], and other supramolecular aggregates, either in consistent environments or adapting to environmental changes (Figure 3).

#### 2.4.1. Nanofiber Assemblies

Among common structures in peptide amphiphilic molecular assemblies are fibrous assemblies. The hydrophobic tail serves as the core, and the peptide portion forms hydrogen bonds to further stabilize the assemblies formed. Functional peptide fragments on the surface provide the assemblies’ functionality. Hartgerink et al. [51] prepared fibrous assemblies with β-sheet structures by using the adsorption of phosphatidylserine with Ca^2+^ and combining it with RGD sequences and C_16_ hydrophobic chain tails that have cell adhesion properties, inducing hydroxyapatite formation akin to collagen fibers’ growth. Utilizing self-assembly’s dynamic nature, they achieved controlled growth through disulfide bond cross-linking. Garcia et al. [56] designed a catalytically active, smallest peptide-based hydrogel, FFH, forming β-sheet structured fiber bundles under physiological conditions.

#### 2.4.2. Nanotube Assemblies

Nanotubes are usually formed from cyclic peptides or peptide amphiphilic molecules. The cyclic peptide itself is a closed-state structure where the molecules form hydrogen bonds on the axial side and thus assemble into nanotubes [57]. Nanotubes have more opportunities for the functionalization of their surfaces than nanofibers, and the presence of hollow structures represents a new way of delivering drugs to specific sites [58]. Due to the size of the nanoscale, some small molecules can be easily encapsulated inside and interact with the cell membrane to enter into the membrane to release the small molecules. Porter et al. [59] prepared nanotubes using FF dipeptide, which is the smallest peptide structure found by humans that can form nanotubes. Also, the authors prepared enantiomers of FF and demonstrated that nanotubes have a better ability to deliver hydrophilic fluorescein, extending the ability of nanotubes to deliver drugs across biological barriers. Rho et al. [60] prepared nanotubes using cyclic peptides and demonstrated a stable, biocompatible, and dynamic nanotube structure by fluorescence resonance energy transfer. The stability of the nanotube assemblies could be modulated by introducing a secondary hydrophobic driving force, and functionalization of the nanotube surfaces did not affect their biological activity.

#### 2.4.3. Nanosheet Assemblies

Compared with the first two assemblies, nanosheets are two-dimensional structures, which are usually prepared from α-helix blocks due to the tendency of peptide amphiphiles to form axial intermolecular hydrogen bonds. With the help of surface functionalization of nanosheets, they are widely used in sensing, catalysis, etc. Insua and Montenegro [61] developed a cyclic heptapeptide, forming two-dimensional nanosheets, a significant advancement in 2D hollow supramolecular materials. Merg et al. [62] used α-helix folded blocks to create controllably sized square nanosheets with growth correlated to monomer addition.

#### 2.4.4. Nanosphere Assemblies

Peptide amphiphilic molecules, resembling liposome structures, can form spherical assemblies like micelles and vesicles. Vesicles, useful in biomedical applications, including drug delivery carriers, artificial cells, and microbial reactors [63,64,65], face challenges due to structural instability [66]. To enhance stability, introducing bioactive recombinant proteins is a new research focus. Booth et al. [67] developed peptide fibrous structures with dynamic covalent bonding, guiding functional behavior post-assembly, which provides a guide to the functional behavior of the minimal peptide after the formation of the assemblies. Jang et al. [68] produced thermoresponsive vesicles. The increase in temperature could transform the vesicles from small monolayer vesicles to large bilayer vesicles; the use of the cavity structure with the hydrophobic structure in the middle of the bilayer could transport molecules with different polarities, which provided key information for the design of globular protein vesicles with specific sizes and membrane structures. Meanwhile, Yin et al. [69] introduced a double hydrophobic chain at the end of the peptide and prepared a spherical micellar assembly that could transport drugs with the help of the π-π stacking effect while increasing the affinity with cells.

### 2.5. Advantages of Self-Assembled Peptides

Peptide molecules, through multiple non-covalent bonding interactions, form a variety of rich nanostructures. These self-assembled peptide nanomaterials offer unique advantages that are not available in general polymer materials, including the following.

#### 2.5.1. Environmental Responsiveness

The process of peptide self-assembly is the result of the synergistic action of multiple non-covalent forces. These changes can affect self-assembly, leading to structural alterations. A series of changes in the self-assembly structure will also occur. The environmental factors that have a greater impact on the self-assembly of peptides include light, enzyme, pH, temperature, and so on. Leveraging this responsiveness, researchers can control the self-assembly and disassembly of peptides, allowing adaptive modifications to different environmental conditions and developing a range of environmentally responsive peptide self-assembly nanomaterials.

#### 2.5.2. Adjustability of Assembly Structure

Peptide molecules, composed by condensing various amino acids, exhibit an array of structures and types. By changing the type and number of amino acids, different types and structures of peptides can be composed. The diversity of the self-assembly module’s biological function modules provides a range of choices for the design of self-assembled peptide materials, leading to the formation of peptide nanomaterials with diverse structures and functions.

#### 2.5.3. Reversibility

Self-assembled peptide structures, once disrupted, can spontaneously reassemble into their original supramolecular structure, and the process does not generate energy consumption. This is particularly evident in enzyme-triggered self-assembly. The hydrogel structure formed by peptide self-assembly is destroyed and then restored under the action of phosphatase and diphosphatase [49]. This reversible nature allows for tailored biological applications through precise control of the peptide self-assembly process.

#### 2.5.4. Suitable Histocompatibility

Peptide self-assembled nanomaterials, compared to other polymeric materials, exhibit low cytotoxicity, high biological compatibility, and are easily degradable. Their favorable histocompatibility broadens their application scope, including the formation of hydrogels and other nanostructures for cell culture scaffolds, tissue repair, and regeneration, with minimal rejection.

### 2.6. Application of Self-Assembled Peptides for Drug Delivery

Recent findings suggest self-assembled peptides, due to their biocompatibility, biodegradability, and multifunctionality, serve as highly efficient drug delivery systems. These systems enhance drug bioavailability, increase chemotherapeutic selectivity, extend drug duration, and significantly improve the physicochemical properties and biological activities of drugs [70].

Self-assembled peptides are extensively explored in tumor therapy. For example, the amphiphilic peptide KLAK induces apoptosis by cleaving mitochondrial membranes [71]. Variants based on this peptide have been developed to combat various cancers. Standley et al. designed the amphiphilic molecule C16-A4G3(KLAKLAK)_2_ by linking KLAK to lauric acid, which can self-assemble into cylindrical nanofibers and disrupt cell membranes (Figure 4A) [72]. KLD12 forms β-sheet gel structures with an alternating distribution of hydrophobic and ionic hydrophilic residues and can form β-sheet gel structures in aqueous solution [73]. Law et al. introduced enzymatic cleavage sites based on KLD12 and designed self-assembling peptides with protease responsiveness, which, when used for drug delivery, can release the drug carried by enzyme-stimulated responsiveness [74]. The peptide RADA16 has the structural feature of a regular distribution of positively charged residues (R) and negatively charged residues (D) separated by hydrophobic residues (A). The hydrogel formed using this peptide contains 99% water and has a significant inhibitory effect on MCF-7 of breast cancer cells when used to deliver tamoxifen, as well as suitable biocompatibility to promote the growth of normal breast cells [75].

Peptides with triggered self-assembly capabilities can induce morphological changes in the tumor microenvironment, thereby enhancing drug activity and reducing toxicity. For instance, micelle-like nanoparticles created by Callmann et al. transform in response to MMPs. Callmann et al. prepared micelle-like nanoparticles by diblock copolymerizing a section of an MMP-substrate peptide with paclitaxel, where the paclitaxel formed a hydrophobic core, and the peptide formed a hydrophilic shell. Upon interaction with MMP, the peptide shell is cleaved and allows the transformation of the nano micelles into micron-sized self-assemblies, a transformation that corresponds to tumor-directed drug targeting, resulting in a significant reduction in systemic toxicity without compromising activity [76]. Moyer et al.’s peptide H6 undergoes pH-responsive structural transformations, forming fibrous self-assemblies at pH 7.5 but converting to nanoparticles as the pH is lowered, thus both prolonging the in vivo stability of the peptide and facilitating peptide uptake at the tumor site (Figure 4B). A higher therapeutic efficiency was demonstrated compared to peptides without responsive structural changes [77]. Self-assembled H6-based nanofibers were used to encapsulate and deliver camptothecin with an encapsulation rate of 60%, which is a 7-fold improvement compared to control nano micellar materials [78]. For drug-resistant types of cancer, in situ-triggered self-assembled peptides also showed enhanced activity and toxicity. Li and colleagues designed peptides with ester bonds responsive to carboxylic acid lipase, enhancing activity against drug-resistant ovarian cancers and reducing systemic burden [79].

Extracellular self-assembly of peptides can inhibit cancer cell migration and induce apoptosis. Kuang et al. designed a D-type tripeptide as a precursor, which, when applied to HeLa cells, formed nanofibers that inhibited cell migration and later activated caspase-3 and caused apoptosis [80]. The team also designed a nuclear peptide that can be dephosphorylated by the extracellular enzyme CD73, forming a gel to inhibit HepG2 cells [81]. Tanaka et al. designed a lipopeptide that could be cleaved by MMP-7 in the tumor microenvironment, thereby forming a gel, which showed significant toxicity to five different cancer cells but low toxicity to normal cells (Figure 4C) [82].

Intracellular self-assembly of peptides shows promise in antitumor drug development. Short peptides by Kuang et al. form nanofibers inside cells, disrupting cytoplasmic protein dynamics, which can cause the death of HeLa cells. Because the self-assemblers are nanoscale and similar to natural protein aggregates in the cytoplasm, the action is restricted to cytoplasmic proteins. Protein profiling and protein blotting results show that aggregates can react with a variety of cytoplasmic proteins (microtubulin, waveform protein and actin, etc.), which will reduce the polymerization of microtubulin, thus hindering microtubule formation, disrupt the dynamics of actin microfilaments and waveform protein intermediate filaments in the cell [80]. Jeena et al.’s Mito-FF aggregates on mitochondria in cancer cells, consisting of diphenylalanine, a mitochondria-targeting fragment, and a fluorescent probe, which selectively aggregates on mitochondria in cancer cells (due to the high negative potential on the membrane), forming fiber-like self-assemblies to cleave the mitochondrial membrane and activate the apoptotic pathway in cancer cells (Figure 4D) [83].

**Figure 4 pharmaceutics-16-00240-f004:**
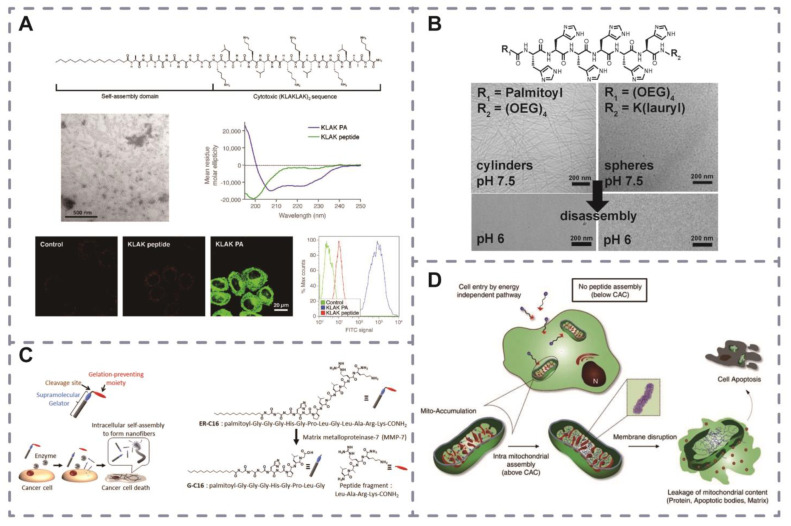
(**A**) KLAK PA self-assembles into nanofibers that stabilize the α-helical peptide conformation and are cell permeable. Reprinted with permission from Ref. [72]. (**B**) A pH-sensitive self-assembled system was developed to both control nanostructure shape and respond to the acidic microenvironment of tumors using self-assembling peptide amphiphiles (PAs). Reprinted with permission from Ref. [77]. (**C**) Cancer cell death induced by molecular self-assembly of an enzyme-responsive supramolecular gelator. Reprinted with permission from Ref. [82]. (**D**) Intra-mitochondrial assembly of Mito-FF. The self-assembly process is driven by the increased mitochondrial membrane potential of cancer cells, leading to high mitochondrial accumulation of Mito-FF, followed by self-assembly into fibrils. The intra-mitochondrial fibrils further disrupt the membrane, resulting in leakage of mitochondrial contents to the cytosol, which eventually induces cellular apoptosis. Reprinted with permission from Ref. [83].

## 3. Functional Peptide-Modified Drug Delivery Systems

### 3.1. Classification of Functional Peptides

#### 3.1.1. Cell-Penetrating Peptide (CPP)

Cell-penetrating peptides (CPPs), typically comprising fewer than 30 amino acids, are primarily basic, amphiphilic, and positively charged and can transport drugs directly through the phospholipid bilayer of cell membranes into cells [84]. One category of CPPs consists of hydrophilic sequences rich in positively charged amino acids (e.g., lysine and arginine), which interact with negatively charged cell membranes to facilitate cytoplasmic entry [85]. Another category includes amphiphilic or hydrophobic sequences that interact with hydrophobic regions of the cell membrane for lipid membrane traversal [86]. The first CPP, derived from HIV-1’s Tat protein (RKKKKRRQRRR), was reported in 1994 by Fawell et al. To date, a variety of major cell-penetrating peptides have been identified, including Tat [87], MAP [88], Transport [89], VP22 [90], Arg/Lys sequence-rich peptides [91], and signal transduction peptides [92].

#### 3.1.2. Cell-Targeting Peptides (CTPs)

The molecular expression status on the surface of tumor cells and tumor vascular endothelial cells differs significantly from that of normal cells, enabling differentiation between tumor and normal tissues from normal tissues. Techniques like phage display, One-bead One-compound (OBOC) libraries, and Positional Scanning-synthetic Peptide Combinatorial Libraries (PSSPCLs) are used to obtain a large number of targeting peptides. When the cytotoxin binds to the cell-targeting peptide, it can be selectively transported and enriched in the tumor tissue, thereby increasing drug concentration in the pathological site for effective tumor eradication while minimizing the impact on normal cells with low expression or non-expression of these receptors, and thus effectively reduce the systemic toxicity of the drug.

Targeting tumor cells is one of the most direct drug delivery strategies. Many primary and metastatic tumor cells overexpress regulatory peptides, such as human epidermal growth factor receptor (HER-2), gonadotropin-releasing hormone receptor (GnRH-R), transferrin receptor (TfR), etc., to varying degrees. Targeted peptides directed at these receptors on the surface of the tumor cells can function as drug delivery warheads. Tumor growth is largely blood supply dependent. Neovascularization not only fosters tumor growth but also facilitates metastasis. Unlike frequently mutating tumor cells, the tumor vascular system is genetically more stable and makes anti-angiogenic therapies less prone to drug resistance [93]. Therefore, directly targeting the neovasculature of solid tumors is also clearly a highly promising mode of drug delivery. Tumor neovasculature endothelial cells tend to overexpress specific proteins, endothelial cell growth receptors, proteases, and cell surface proteoglycans, among other proteins [94]. The best-known tripeptide motifs targeting the tumor vasculature are arginine-glycine-aspartic acid (RGD) and asparagine-glycine-arginine (NGR).

#### 3.1.3. Stimulus-Responsive Peptide

Peptides can respond to environmental stimuli, originating either internally, such as pH differences or enzyme overexpression [95], or externally, like light-induced temperature changes [96]. pH-responsive peptide sequences exhibit conformational changes or charge alterations in response to pH changes [97], targeting the acidic environments of most solid tumors (pH = 6.5–7.2) and lysosomes/endosomes (pH = 4.0–5.0). Temperature influences peptide conformation and solubility [98]. Elastin-like peptides (ELPs) exhibit significant solubility shifts at specific temperatures [99], which have been applied to various drug delivery systems.

### 3.2. Peptide-Modified Nanomaterials

Conventional nanomaterials, including inorganic (e.g., quantum dots, metal nanoparticles, mesoporous silica nanoparticles), organic (e.g., semiconducting and dendritic polymers), and organic–inorganic hybrid (e.g., metal–organic frameworks, polymer-coated inorganic nanoparticles) varieties. Once these are modified with a variety of different functional peptides to form multifunctional nanocarriers, the advantages of peptide-based nanomaterials will be further amplified with the multifunctional features of peptides, improving the cellular uptake of drugs [100,101].

#### 3.2.1. Peptide-Modified Liposomes

Liposomes are microscopic vesicles formed by lipids like phospholipids. With the advantages of low toxicity, biodegradability, and the ability to alter the pharmacokinetic properties of encapsulated drugs, liposomes have attracted extensive attention in the field of drug delivery. However, liposomes lack tissue-cell specificity, so they have been modified with peptides and other ligands to achieve tumor-targeted drug delivery, improve drug efficacy, and reduce toxic side effects.

Fu et al. [102] prepared cyclic RGD (cRGD)-modified liposomes (LPs) (cRGD/CD) based on an independently synthesized tripeptide head of lipids (CDO14) and DOPE and loaded with doxorubicin (DOX) for targeted lung cancer treatment (Figure 5A). These liposomes exhibited increased cellular uptake due to the targeted action of cRGD and the acid instability of CDO14, facilitating lysosomal escape and drug release. The results of both in vivo and ex vivo studies showed that cRGD/CD-DOX had stronger antitumor activity compared with the control group.

Similarly, NGR peptide-modified liposomes, consisting of asparagine (N)-glycine (G)-arginine (R), bind specifically to aminopeptidase N (CD13) in the membranes of tumor cells and have therefore been used for targeted modification of LPs [107]. Gu et al. [108] inserted NGR peptide onto the surface of pH-sensitive LPs encapsulated with paclitaxel (PTX) and used HT-1080 human fibrosarcoma cells with high expression of CD13 as the experimental target, and the experimental results confirmed that NGR peptide-modified LPs could specifically target HT-1080 cells, and significantly increased the antitumor activity of LPs.

Kuai et al. [109] designed a controlled LPs modified by polyethylene glycol (PEG) and Tat. Among them, PEG is connected by disulfide bonds, which can protectively mask Tat, and when the complete cleavage agent L-cysteine (L-Cys) is added, the disulfide bonds can be broken, releasing the spatial resistance of PEG. This exposes the “functional molecule” Tat, which can increase the cellular uptake of LPs and thus achieve cellular control of LPs uptake. Due to the lack of targeting of CPP itself, LPs modified with CPP alone often fail to target tumors. To address this issue, Shi et al. [110] designed a peptide with the functions of active targeting of integrin αvβ3 and cell penetration, a peptide that actively targets integrin αvβ3 and enhances cell penetration. The results of the study showed that the tandem peptide-modified paclitaxel LPs achieved a tumor suppression rate of 85.04% in hormonal mice, and the survival rate of the mice was significantly higher than that of all other control groups.

Deng et al. [111] prepared internalizing RGD (iRGD)-modified LPs (iRGD-LTSL-DOX) loaded with DOX. These LPs took advantage of the targeting and cell-penetrating properties of iRGD to bring the drug carrier into the tumor cells and release DOX for tumor cell killing. The in vitro experiments compared the targeting ability of cells with different degrees of αvβ3 expression, and the results showed that the targeting ability of iRGD-LTSL-DOX was 1.8~2 times higher than that of LTSL-DOX in cells with high expression of αvβ3. This result suggests that iRGD-modified LPs can effectively target and achieve tumor cell killing. Yu et al. [112] used spatially stable LPs modified with iRGD (iRGD-SSL-DOX) for melanoma treatment. Melanoma cells B16-F10 overexpressing αv integrin receptor recognized iRGD-modified LPs. In vitro as well as in vivo experimental results showed that iRGD-modified LPs had higher tumor-targeting and tumor-penetrating activities. In vivo, antitumor results showed that iRGD-SSL-DOX had stronger anti-melanoma effects than SSL-DOX in B16-F10 ruffled mice.

#### 3.2.2. Peptide-Modified Polymers

Polymers are favored for their high molecular weight, stability, and ease of chemical modification. Researchers have prepared a variety of cancer-targeted polymeric drug delivery systems by modifying specific ligands or synthesizing tumor environment-sensitive materials.

Based on the high affinity of LFC131 peptide for X-C receptor type 4 overexpressed in hepatocellular carcinoma, Sun et al. [103] and others delivered epirubicin with LFC131-modified poly (lactic acid hydroxyacetic acid) copolymer (Figure 5B). In vitro and in vivo experiments showed that the LFC131 peptide-functionalized delivery system resulted in approximately 3-fold higher HepG2 uptake than non-targeted cells and was effective in inhibiting hepatic tumor growth with a significant reduction in systemic side effects. Kathrin et al. [113] used peptide GE11 as a targeting ligand and polymer synthesized with polyethyleneimine (LPEI) and PEG as a carrier to prepare LPEI-PEG-GE11/NIS, a drug delivery system targeting hepatocellular carcinoma cells Huh7. LPEI-PEG-GE11/NIS delivered 22 times more ^131^I into Huh7 cells than the non-GE11-modified delivery system and showed a better radiotherapy effect on liver cancer.

A54 peptide is a liver cancer-targeting peptide discovered by Du et al. [114], which is more abundant in tumor tissues than in normal liver tissues. Based on this, Situ et al. [104] synthesized the A54-poly (lactic acid hydroxyacetic acid) copolymer (PLGA)-dexamethasone (DEX) complex particles A54-PLGA-DEX, which showed a 75% encapsulation rate and was able to target hepatocellular carcinoma cells BEL-7402 and HepG2 (Figure 5C).

Wang et al. [115] constructed the drug delivery system RGD-SRF-QT using RGD-modified PLGA polymers encapsulated with lipids and co-delivered sorafenib (SRF) and quercetin (QT) to hepatocellular carcinoma, which significantly inhibited the growth of liver tumors. Wu et al. [116] also used RGD-modified superparamagnetic iron oxide functionalized polyethene glycol grafted polyetherimide hepatocellular carcinoma-targeted delivery of siRNA, and the results demonstrated that the RGD-modified gene delivery system showed higher gene transfection efficiency and better apoptosis ability for Bel-7402. Zhang et al. [117] developed transferrin (Tf)-modified self-assembled PLGA nanoparticles for co-delivery of DOX and cisplatin (DDP) and found that Tf-DOX/DDP polymeric nanoparticles showed higher cytotoxicity and stronger antitumor activity than the unmodified Tf-based single-drug delivery system, which enhanced the synergistic efficacy of the two drugs in the treatment of liver cancer.

#### 3.2.3. Peptide-Modified Inorganic Nanoparticles

Inorganic nanoparticles have a small particle size, large surface area, high adhesion to biofilm, favorable to local retention, and easy to be phagocytosed by the reticuloendothelial system or mononuclear phagocytosis system after oral or injectable administration and binding with plasma proteins, which makes them easy to be aggregated in the mononuclear macrophage-rich organs.

The advantages of gold nanoparticles, for instance, have been employed for both diagnostic and therapeutic purposes in tumor treatment. Huang et al. [105] found that adriamycin-loaded gold nanoparticles, modified with A54 peptide, significantly enhanced antitumor effects (Figure 5D). In another study, gold nanoparticles modified with a 12-amino-acid peptide were used for the photothermal treatment of hepatocellular carcinoma cells, and the results showed that the peptide-functionalized gold nanoparticles exhibited suitable targeting of hepatocellular carcinoma cells, BEL-7404 and BEL-7402, and induced the death of the cancer cells more efficiently after treatment with near-infrared light [118]. Recently, the apoptosis-inducing and anticancer effects of pro-apoptotic peptide-modified gold nanoparticles were also found to be significantly enhanced compared with those of free peptide-modified gold nanoparticles [119].

Liao et al. [106] first synthesized novel microspheres consisting of silica and organoalginate and then modified doxorubicin-loaded microspheres with RGD peptides and found that RGD-modified drug-loaded microspheres had 3.5-fold higher drug delivery efficiency within HepG2 than those unmodified with RGD (Figure 5E). Bijukumar et al. [120] found that the uptake of transferrin-coupled biodegradable graphene (Tf-G) in hepatocellular carcinoma cells was significantly increased compared with that of bare graphene, removed for brevity and relevance to the overall topic.

### 3.3. Peptide–Drug Conjugate

Peptide–drug conjugates (PDCs) are synthesized by covalently linking drugs to peptides through bonds (Figure 6), serving as prodrugs, enhancing the bioavailability of hard-to-dissolve drugs, enhancing the selectivity of chemotherapeutic drugs, and prolonging the duration of drug action, which can significantly improve the physicochemical and biopharmacological properties of drugs [121,122]. Research indicates that PDCs offer five key advantages over antibody macromolecules when targeting peptide–toxin conjugates [123,124,125]:

(1)Their small molecular size (2–20 kDa) facilitates easier penetration into the tumor stroma and cells;(2)PDCs are not subject to FcR, RES, or ADA pathway-mediated non-pharmacological elimination, which effectively improves drug utilization;(3)PDCs can be produced in prokaryotic nuclei or through chemical synthesis, simplifying production and scalability;(4)PDCs can be conjugated with clinically proven cytotoxic agents like Adriamycin and Paclitaxel, which are proven cytotoxic molecules, to prepare target agents, significantly reducing off-target toxicity and enhancing PDC formulation platform technology’s feasibility;(5)Certain targeted peptides in PDCs can alter cell entry mechanisms to effectively kill drug-resistant tumors, addressing the challenge of traditional chemotherapy’s ineffectiveness against drug-resistant tumors.

#### 3.3.1. Linker in the Peptide–Drug Conjugate

Linkers are intermediates that covalently couple cytotoxic molecules to peptides. The advantage of PDC is that it maintains the prodrug state during somatic circulation and releases the antineoplastic drug only at the tumor tissue target. Thus, the linker plays a crucial role in this sequential delivery-release process. To achieve this desired tumor-targeted killing function, the linker must satisfy several core requirements as follows [126,127]:

(1)Suitable stability during the somatic circulation before reaching the target site to avoid systemic toxicity caused by drug release at non-pathological sites;(2)After being phagocytosed by target cells, PDC can be triggered by the special microenvironment within target cells to rapidly break off and release drug molecules;(3)The hydrophobicity should not be too strong; otherwise, PDCs are prone to poor in vivo stability and decreased drug efficacy due to hydrophobic aggregation and, at the same time, produce strong systemic toxicity and immune side effects.

According to the release mechanism and circulatory stability of different types of drugs, linkers are mainly divided into two categories: cleavable and non-cleavable. According to the different triggering mechanisms of the tumor microenvironment, such as high concentration of glutathione, low pH, and high concentration of specific proteases, cleavable linkers can be further classified into chemically cleavable and enzyme-sensitive types [128]. Linkers with different cleavage mechanisms can directly affect the half-life of PDC drugs [129].

##### Non-Cleavable Linkers

Non-cleavable linkers are those that connect the tumor-targeting peptide to the drug portion through non-cleavable bonds such as amide and ether bonds. Peptide-coupled drugs using this type of linker usually reach the target tissue without cleavage of the linker and without affecting the cytotoxicity of the drug [130]. Common non-cleavable linkers include 6-aminohexanoic acid, short peptide fragments such as CGGW, and membrane-penetrating peptides like Tat. These are typically peptides consisting of four amino acids or the main chain of a carbon chain containing 5–8 carbon atoms [130,131,132], chemically stable, and able to regulate the polarity of tumor-targeting peptide-coupled drugs [133,134].

##### Cleavable Linker

pH-sensitive linker: Tumor cells proliferate and metabolize more rapidly than normal tissue cells, resulting in the lactic acid build-up in the tumor cells, leading to a pH value in the tumor microenvironment of around 6.8, whereas the pH value of the blood in the human body is around 7.3. pH-sensitive linkers are designed to take advantage of this change in pH value [135]. Tumor-targeting peptide-pH-sensitive linker-drug coupling does not cleave when it enters the human circulatory system but cleaves to release the drug in the acidic environment when it reaches the tumor tissue. Langer et al. [136] showed that neuropeptide Y-hydrazone-donoramycin, obtained by coupling neuropeptide Y (NPY) to donoramycin using an acid-sensitive hydrazone bond, was significantly more cytotoxic than neuropeptide Y-amidodonoramycin, which was coupled using an amide bond. Other linkers that are susceptible to hydrolysis in weak acidic environments are the enol ether bonding group and the imide group [137,138], which are also commonly used for linking drugs to macromolecular compounds.

Enzyme-sensitive linker: Enzyme-sensitive linkers remain stable in the human circulatory system but undergo specific enzymatic cleavage when they reach sites rich in target enzymes. Histone B is highly expressed in a variety of malignant tumors and is closely associated with tumor invasion and metastasis [139]. Chen et al. [139] found that a linker containing the short peptide sequence GFLG could be specifically cleaved by histone B, releasing doxorubicin in tumor cells. MMP is a family of proteases that can target the extracellular matrix, and among them, MMP2 and MMP9 are highly expressed in rectal carcinoma, neuroblastoma, bladder cancer, and hepatocellular carcinoma [140,141,142,143]. MMP2 and MMP9 play an important role in tumor invasion and metastasis by degrading fibrillar collagen after cleavage by collagenase [142]. The short peptide sequence PLGLAG is an MMP2/MMP9-sensitive linker fragment that breaks in tumor tissue [144].

Oxido-reduced linker: Reduced glutathione (GSH) is a small cysteine-containing peptide in the body that is found in significantly higher concentrations in tissues such as lung, pancreatic, head and neck, and breast cancer than in normal human tissues. Glutathione-sensitive linker groups are coupled to drugs by forming disulfide bonds, and when drugs containing such linkers reach the tumor tissues, the linkers are cleaved by glutathione, and the cytotoxic load exerts its potency. Song et al. [145] showed that low pH insertion of peptide–sulfur–sulfur–doxorubicin (pHLIP-SS-DOX) can target acidic tumor cells and reverse multidrug-resistant proteins. Meanwhile, GSH-mediated cytotoxicity studies in vitro showed that pHLIP-SS-DOX had significant cytotoxicity at pH 6.0.

#### 3.3.2. Applications of Peptide–Drug Conjugate

Numerous membrane-penetrating peptides have been effectively utilized to enhance the cell entry efficiency of proteins, nucleic acids, and small molecules. For example, leveraging positively charged cell-penetrating peptides to link with negatively charged nucleic acid macromolecules to construct nano-delivery systems for the treatment of various genetic disorders enhances the in vivo stability and transmembrane capacity of nucleic acid macromolecules. Duan et al. [146] designed a paclitaxel membrane-penetrating peptide conjugate with strong membrane-penetrating and antitumor activity. This conjugate not only amplified paclitaxel’s antitumor effectiveness but also addressed its poor solubility of paclitaxel and tumor drug resistance.

By attaching peptides with specific targeting properties to drugs, highly targeted drug delivery systems can be engineered. For example, the prevalent overexpression of the LHRH receptor in certain cancers offers an opportunity to direct drugs specifically to cancer cells using LHRH-targeting peptides. Combining DOX and 2-pyrrolyl-DOX with LHRH-targeting peptides increases drug targeting to cancer cells and reduces drug toxicity. Lindgren et al. [147] attached YTA2 and YTA4 peptides to methotrexate to form a covalent complex that significantly increased its potency against the target enzyme dihydrofolate reductase 15–20-fold compared to free methotrexate. Intercellular adhesion molecule-1 (ICAM-1) peptide sequence derived from cell surface proteins can be specifically endocytosed by leukemia T-cells. Linking ICAM-1 peptide to therapeutic drugs can improve the therapeutic effect of drugs on cancer and autoimmune diseases and reduce the toxic side effects of chemotherapy.

Nishimura et al. [148] reported a GALA peptide that promotes lysosomal escape, which is an amphiphilic, pH-sensitive peptide with membrane fusion capabilities when it reaches the weakly acidic microenvironment of the tumor. Matrix metalloproteinases (MMPs) are a class of tumor markers that are highly expressed at the site of invasive tumors and at lower levels in normal cells. Zhu et al. [149,150] investigated MMP-2-sensitive octapeptide for controlled drug release at tumor sites, reducing non-target tissue exposure and enhancing therapeutic outcomes. Tu et al. [151] designed a stimuli-responsive self-assembled multifunctional polymer PEG-ppTat-DOX, where Tat is a cell-penetrating peptide, and designed a stimuli-responsive, self-assembling multifunctional polymer PEG-ppTat-DOX. This nano-complex efficiently targets antitumor drug delivery and augments chemotherapeutic effects.

#### 3.3.3. Advances in Clinical Studies of Peptide–Drug Conjugate

Currently, only two peptide–drug conjugates (PDCs) are FDA-approved for cancer therapy: melflufen (Pepaxto^®^) and ^177^Lu-dotatate (lutathera^®^) (Table 2).

Melflufen is approved in combination with dexamethasone for the treatment of patients with severe relapsed or refractory multiple myeloma (R/R MM) [152]. The FDA’s accelerated approval of melflufen was based on the results of the Phase II HORIZON study in heavily treated, drug-resistant, and high-risk R/RMM patients. In addition, results from the phase III OCEAN study (NCT03151811) demonstrated longer progression-free survival (PFS) with melflufen plus dexamethasone compared to standard treatment with pomalidomide plus dexamethasone in patients with RRMM. However, melflufen was withdrawn due to safety concerns.

The approval of ^177^Lu-dotatate was based on the results of the phase III NETTER-1 study [153]. This study demonstrated that ^177^Lu-dotatate given once every 8 weeks (four total doses) in combination with octreotide (LAR) in patients with metastatic midgut neuroendocrine tumors provided a significant advantage in terms of PFS, ORR, and OS compared to LAR alone.

The exploration and application of novel targeted peptides have led to numerous PDCs entering clinical research for various solid tumors, including breast cancer and non-small cell lung cancer. The most advanced in development is represented by paclitaxel (SNG1005), co-developed by AngioChem and Shenogen Pharma Group, and zoptarelin doxorubicin (AEZS-108), co-developed by Aeterna Zentaris and Sinopharm Yixin. SNG1005, a brain-targeting peptide–drug conjugate, has shown positive clinical efficacy in treating brain metastases from breast cancer. Phase II clinical results showed positive clinical efficacy in the treatment of molluscum contagiosum metastatic carcinoma of the brain from breast cancer and parenchymal metastatic carcinoma of the brain from recurrent breast cancer. AEZS-108 is currently in trials for desmoplasia-resistant prostate cancer. Other PDCs in development suggest both potential and challenges in translating PDCs’ pharmacodynamics into clinical success. To date, the results of these PDCs have been mixed, suggesting that several challenges remain in translating the favorable pharmacodynamic properties of PDCs into improved clinical outcomes for patients.

## 4. Strategies to Improve Peptide Drug Delivery

Peptide drugs are recognized for their low toxicity, low immunogenicity, high tissue permeability, and easy synthesis and modification [154]. This results in poor in vivo stability by a variety of proteases, high blood and renal clearance, and many limitations in practical applications [155,156]. To address these issues, researchers have employed various strategies to improve the stability of peptides involved in circulation in organisms.

### 4.1. Modification of Peptide Structure

Peptides exhibit instability in organisms due to their special molecular structure. The main chain of peptides is prone to deamidation, especially the amide group located on the surface of the molecule, which is easy to recognize and hydrolyzed by protease. Amino acid residues on the side chain are prone to conformational changes, such as the dissociation of α-helix and the breaking of salt bridges. All these factors lead to the fact that peptides are not stable in the body for a long period. To solve the structural instability of peptides, many researchers have been devoted to the structural modification of peptides [157]. Approaches include the cyclization of peptides, substituting L-amino acids with the activity of the peptide structure to participate in the degradation reaction, such as connecting the peptide head to tail or side chain to side chain to form a cyclic peptide, replacing natural L-amino acids with D-amino acids that are not easily recognized by proteases, and altering the sequence of amino acids in the peptide.

#### 4.1.1. Cyclic Peptide Formation

Cyclic peptides are more structurally stable than linear peptides [158]. The latest new drug to receive FDA regulatory approval, Rezafungin, a peptide analog of echinocandin for the treatment of candidemia and invasive candidiasis, has gained FDA approval in drugs like Rezafungin. The formation of cyclic peptides by linking peptides head to tail through disulfide bonds or the use of cysteines to form small intramolecular loops. It deprives the peptide chain of its free amino and carboxyl groups and prevents it from being degraded by proteases that specifically recognize the end-group amino and carboxyl groups. Moreover, cyclic peptides form disulfide bonds or salt bridges within the molecule, and these interactions further enhance the stability of the peptide. Bogdanowich-Knipp et al. [159] compared the stability of linear RGDs with cyclic RGDs and found that aspartic acid is the degradation site of RGD peptides and that its side chains readily undergo intramolecular reactions with either the C-terminus or N-terminus of the peptide chain. When the RGD is linear, the molecular flexibility is larger, resulting in a shorter distance between the two atoms involved in the reaction, and the degradation reaction is easy to occur. When the RGD is cyclic, the rigidity of the molecular structure is enhanced, resulting in an increased distance between the atoms involved in the reaction, which prevents the degradation reaction from occurring. Another reason for the increased stability of cyclic RGDs is the formation of a salt bridge between the guanidine group in arginine and the carboxyl group of aspartic acid: alpha or beta. This interaction also prevents the side chain of aspartic acid from participating in the degradation reaction.

Gomesin, an 18-amino-acid peptide, exhibits various biological activities with the sequence ZCRRLCYKQRCVTYCRGR, where Z is pyroglutamic acid. This peptide chain has antitumor, antipathogenic microbial, antifungal, antileishmanial, and antimalarial properties. To improve the stability of Gomesin in pepsin, Chan et al. [160] replaced the N-terminal pyroglutamic acid in Gomesin with glycine and then connected the first and last parts of Gomesin to form a cyclic peptide. It was found that when the cyclic peptide was folded, a disulfide bond was formed between the cysteines to make the peptide structure more stable, and the toxicity of the cyclized Gomesin to HeLa cells was increased by 3-fold, and the antiplasmodial and antibacterial activities of the cyclized Gomesin were also significantly improved. Hymenochirin-1B, an α-helical peptide, achieved stability through that has a wide range of biological activities, such as antipathogenic microorganisms, anticancer, immunomodulatory, and antidiabetic activities. Natural α-helical linear peptides usually fail to maintain their expected conformation and binding ability to their intended targets. Li et al. [161] achieved the stabilization of the peptide hymenochirin-1B by covalently linking the side chains of two amino acids using ruthenium-catalyzed olefinic complexation, resulting in the formation of an intramolecular mini-cycle of the peptide. The researchers synthesized a total of 10 hymenochirin-1B peptide analogs and examined their stability in the trypsin environment. It was found that all the sequences containing cyclic peptides showed better protease stability than the original sequence of hymenochirin-1B. Among them, H-10 was the most protease resistant, with a half-life of 3.5 h. Moreover, H-10 showed better inhibitory activity against tumor cells.

#### 4.1.2. Use of D-Amino Acids

Since the amino acids recognized by proteases in vivo are naturally occurring L-amino acids, researchers have attempted to substitute L-amino acids with protease-resistant D-amino acids [162]. Some researchers replaced only the C- and N-terminal amino acids of the peptide chain with D-amino acids, while others changed all amino acids in the peptide chain to D-amino acids. Tugyi et al. [163] selected a segment of the antigenic determinant cluster (^15^TPTPTGTQTPT^25^) in MUC2 mucin and replaced the three amino acids at the C-terminal and N-terminal ends with D-amino acids or replaced the three amino acids at both ends with D-amino acids, respectively, to study the stability of these amino acids in human serum. It was found that replacing the N-terminal amino acids with D-amino acids (Pep2 and Pep3) greatly improved the stability of the peptides. For example, the degradation rate of Pep2 was 16% and 45% in 10% and 50% serum, respectively. Peptide chains with three D-amino acids at both the N- and C-termini were stable even after 96 h in 50% human serum. Two peptide chains, Pep4 and Pep5, were found to have both high protease stability and a suitable ability to specifically recognize antibodies.

Another study found that the heptapeptide GICP (SSQPFWS), which consists of natural L-amino acids, has a high affinity for the VAV3 protein, which is highly expressed in glioma cells. Due to its unique amino acid sequence, it cannot be subjected to D-amino acid substitutions, but linking it to a D-peptide improves the protease stability of the peptide. ^D^A7R (^D^R^D^P^D^P^D^L^D^W^D^T^D^A) is a ligand for vascular endothelial growth factor receptor-2 (VEGFR2) and neuropilin-1 (NRP-1). Zhang et al. [164] designed to link GICP to ^D^A7R via several glycines to form the peptide conjugate ^D^A7R-GICP (^D^R^D^P^D^P^D^L^D^W^D^T^D^AGGGCGGGSSQPFWS). By comparing the three peptides, GICP, ^D^A7R, and ^D^A7R-GICP, ^D^A7R-GICP was found to be a targeted peptide that did not affect the respective binding sites of the two peptides while improving their protease stability.

#### 4.1.3. Altering Single or Multiple Amino Acids in a Peptide

The main chains of peptide chains depend on electrostatic interactions between the main chains and side chains to maintain structural stability, and the electrostatic interactions between peptide chains can be enhanced by replacing a few of the amino acids. Chen et al. [165] synthesized four peptides by replacing some or all of the lysines in the cleavage peptide PTP-7 with histidine, namely PTP-7 (FLGALFKALSKLL), PTP-7a (FLGALFHALSKLL), PTP-7b (FLGALFKALSHLL), and PTP-7c (FLGALFHALSHLL). The PTP-7c peptide, which replaced both lysines with histidine, was found to have the longest half-life in 100% human serum, 11 h. All histidine-containing peptides were significantly less toxic to normal cells than PTP-7. The histidine-containing peptides PTP-7a and PTP-7b remained active against cancer cells. It can be speculated that the presence of histidine plays a key role in improving the stability of peptides as well as reducing their cytotoxicity.

Structural modification of the N- and C-termini of the peptides can evade recognition by aminopeptidases and carboxypeptidases. Seebac et al. [166] homologated both ends of neurotensin (NT). Insertion of a -CH2 group into the N- and C-termini of the peptide replaced the α-amino acid with a β-amino acid and altered the (R)/(S) configuration of the carbon atom at the center of the amino acid at both ends. It was found that one of the NT analogs (H-(S)-β^2^hR-RPYI-β^3^hL-OH) showed a unique and excellent stability in human serum and remained biologically active in human serum for 7 days.

CTT peptide (CTTHWGFTLC) is a gelatinase inhibitory peptide that was found to have the ability to specifically target tumors [167]. Björklund et al. [168] used alanine to replace amino acid residues in CTT peptides one by one in search of key residues affecting the inhibitory activity of gelatinase and found that tryptophan is essential for gelatinase inhibitory activity. Tryptophan was replaced with tryptophan analogs 5-hydroxytryptophan, 5F-tryptophan, and 6F-tryptophan. It was found that 5F-CTT had a half-life of 3 h in serum and showed a 6-fold higher serum stability than natural CTT. Meanwhile, 5F-CTT also had a better ability to inhibit the migration of tumor cells, and the higher stability of 5F-CTT in serum could be attributed to the aggregation of peptides caused by hydrophobic fluorine atoms, which resulted in a more compact conformation of 5F-CTT.

### 4.2. Modification of Peptide Ends

Modifying peptide ends is a key strategy to protect peptides from protease degradation. Peptide modification with PEG can render peptides “invisible”. Combining peptides with inorganic compounds and self-assembling them into nanoparticles hides the C- and N-termini of the peptide, thus providing resistance to degradation by aminopeptidases and carboxypeptidases.

#### 4.2.1. Hydrophobic Modification of Peptides

Glucagon-like peptide-1 (GLP-1) is a crucial hormone for the treatment of type II diabetes mellitus, but it is susceptible to rapid inactivation by metabolic enzymes such as dipeptidyl peptidase IV (DPP-IV) and neutral endopeptidase (NEP) under physiological conditions [169]. Recent research focuses on the development of long-acting GLP-1 derivatives. Han et al. [170] first introduced cysteine into GLP-1(7-36)-NH_2_ containing glycine at position 8 and then linked multiple aliphatic chains with maleimide to the sulfhydryl group of cysteine to generate multiple aliphatic chain-modified peptide derivatives. It was found that the half-life of G_8_-GLP-1(7-36)-NH_2_ in rat plasma at 37 °C was only about 0.5 h. The half-life of most derivatives was increased. It was further found that the length of the fat chain was closely related to the half-life. Under the same conditions, the half-life of the C8 chain was 5.9 h, that of the C12 chain was 24 h, and that of the C16 chain was 45.4 h. The researchers concluded that the in vitro stability of G_8_-GLP-1(7-36)-NH_2_ derivatives was related to their albumin binding ability, and it was hypothesized that the fatty chains promoted the non-covalent binding between the peptide derivatives and serum albumin, which improved the stability of the peptides.

Nanoparticle construction offers another avenue to improve the bioavailability of peptides. C16Y (DFKLFAVYIKYR) is an integrin-targeting peptide. Ding et al. [171] created an amphiphilic chimeric peptide, DEAP-C16Y. The hydrophobic end of DEAP-C16Y consists of four functional DEAP (3-diethylaminopropyl isothiocyanate) hydrophobic molecules and eight leucine residues, utilizing three lysines to provide the primary amino group attached to DEAP, with C16Y acting as the hydrophilic end, with glycine attached to both ends to remove the terminal charge. At physiological pH conditions, DEAP-C16Y can be assembled into nanomicelles, releasing the peptide at tumor sites. DEAP-C16Y micelles showed better inhibition of angiogenesis, tumor growth, and metastasis than free C16Y peptide, attributed to the hydrophobic DEAP and leucine at the N-terminal end of the C16Y peptide, which enhanced the stability of the C16Y peptide.

#### 4.2.2. Hydrophilic Modification of Peptides

Tumor-associated macrophages (TAMs), particularly M2 macrophages, are potential targets for adjuvant cancer therapy because of their important role in promoting cancer cell proliferation, angiogenesis, and metastasis. The macrophage-targeting peptide (M2pep) specifically binds to M2 cells, but M2pep is not easily effective in vivo due to its poor water solubility. To improve the stability of M2pep in vivo, Ngambenjawong et al. [172] attached M2pep to a polymer composed of hydrophilic and biocompatible N-(2-hydroxypropyl) methacrylamide (HPMA) and N-(3-aminopropyl) methacrylamide (APMA). The researchers attached three M2pep peptide analogs to the polymers to form M2pep-SH, (Ac)M2pep(RY)-SH, and DFBP-cyclized M2pep(RY)-Alkyne, respectively. It was found that upon binding to the polymer, the peptide remained stable in serum even after 24 h, whereas the three free peptide analogs halved in activity within the same time. In particular, the combination of cyclized M2pep(RY) with the polymer retained the targeting properties and increased the stability of the peptide in vivo.

R_8_ is a cell-penetrating peptide that mediates the entry of drugs, nucleic acids, and polymeric nanoparticles, among others, into cells. However, the R_8_ sequence is composed of positively charged amino acids, leading to its easy binding to plasma proteins and poor stability [173]. Cheng et al. [174] linked the KLAK peptide with PEG to protect the peptide from enzymatic degradation, and Wu et al. [175] linked the membrane-penetrating peptide R_8_/cyclic RGD with the phospholipid DSPE via PEG and then embedded it in liposomes loaded with ergosterol (ERG) and cisplatin (DDP). It was found that the half-life of RGD/R_8_-DDP/ERG-LIP in vivo reached 24 h. The reason for such a suitable stability could be that PEG shielded the positive charge of the membrane-penetrating peptide R_8_ and the cyclic RGD was not readily recognized by proteases.

### 4.3. Physical Encapsulation of Peptides

Beyond peptide modification, encapsulation with polymeric substances provides a protective barrier between the peptide and the protease, thus offering enhanced resistance against protease cleavage. Qu et al. [176] developed a nanoscale oral delivery system for oral delivery of GLP-1. The strategy was to attach the peptide to solid nanoparticles of silica and then coat it with a pH-sensitive polymer acrylic resin (SPN-GLP-1). The pH-sensitive acrylic resin effectively preserves GLP-1 stability in acidic conditions such as the stomach, as 80% of GLP-1 was released at pH 7.4, and only about 30% of GLP-1 was released at pH 1.0. For enteric enzyme stability, they tested mixed free GLP-1 and SPN-GLP-1 with intestinal fluid. Due to the presence of high concentrations of digestive enzymes and brush border peptidase, GLP-1 rapidly degraded with a half-life of (2 ± 0.02) minutes. Zhao et al. [177] utilized hydrophobic proteins with self-assembling activity to encapsulate the GLP-1 molecule in the self-assembling cavity, thus preventing GLP-1 from being degraded by proteases.

### 4.4. Novel Technology for Peptides: Microfluidics

As peptide drug research and demand grow, the scale of the peptide industry is growing rapidly, and so does the need for environmentally friendly and safe pharmaceutical processes. The length and complexity of the peptide chain of peptide drugs are gradually increasing, and some of them have more than 30 amino acids. Synthesis challenges are increasing, and the demand for process monitoring and automated synthesis is increasing. Microfluidic technology, recently applied to peptide synthesis, addresses these challenges. Its benefits include in situ peptide cleavage, with fast and short reaction speed; reagents miniaturization reducing hazardous compound handling; the time and temperature can be accurately controlled with high reproducibility; and the capability for automated continuous synthesis and rapid scaling [178]. Wang et al. [179] designed a multi-channel continuous-flow microfluidic chip, drastically shortening synthesis time, reducing the number of reagents, and enabling the simultaneous production of six different peptides with high efficiency and environmental safety. Qiang et al. [180] employed a high-throughput microfluidic chip with a green fluorescent protein color development method to monitor the growth of bacterial cells for expediting new drug molecule screening. Additionally, technologies like ionic liquids are being used, such as ionic liquids that can improve peptide solubility and accelerate the coupling reaction without the need to use a large amount of coupling agent [181].

### 4.5. AI-Enabled Peptide Design and Synthesis

Researchers are exploring computer-based methods, such as virtual screening and molecular docking, to reduce the time and cost of developing new drugs. However, these methods suffer from challenges related to accuracy and efficiency issues. Artificial intelligence, including deep learning and machine learning algorithms, is viewed as a solution to overcome these issues in drug design and discovery. In particular, AI has great potential in peptide drug design, effectively addressing the generating of new peptide molecules through amino acid permutations and combinations. Peptide drug design is further supported by Computer-Aided Peptide Design (CAPD), which integrates rational drug design with computer-aided design technology, providing theoretical guidance through AI, graphic processing, and data mining [182]. 

To address the complexities of peptide synthesis, Mijalis et al. [183] developed automated flow peptide synthesis (AFPS), significantly accelerating the speed of solid-phase synthesis and ensuring precise control over the chemical reaction. The combination of artificial intelligence and automation technology is expected to play a crucial role in expediting drug discovery and chemical synthesis in the future.

## 5. Conclusions and Perspectives

Peptides, known for their excellent physicochemical properties and biocompatibility, have led to the development of self-assembled nanocarriers and functional peptide–drug combinations, marking a significant advance in the construction of intelligent nano-drug delivery systems.

This review includes the progress in peptide-mediated drug delivery systems. Peptide-based drug delivery systems have the advantages of suitable biocompatibility, rich functional groups, intrinsic bioactivity, and specific targeting. Peptides have a wide range of applications in drug delivery systems. To date, research on peptide drug delivery systems has primarily focused on inherent characteristics such as targeting or penetration. However, the development of more sophisticated and precise drug deliveries using responsive and assembled peptides remains an underexplored area. Utilizing in vivo chemical conditions, namely physicochemical stimuli, to modulate drug delivery strategies is gaining traction in research circles.

Recent advancements in peptide solid-phase synthesis, phage display technology, computer simulation, and computation, and more and more new tumor-targeting peptide-coupled drugs have led to the creation of such as the bicyclic peptide drug BT-1718 by Bicycle Therapeutics is a new type of peptide drug with a bicyclic structure [184], which can effectively overcome the problem of fast metabolism of ordinary peptide drugs in vivo. It is anticipated that future developments will leverage these novel peptides to create more stable or multifunctional tumor-targeting drugs.

Solid-phase synthesis remains the primary method for peptide production, but the conventional solid-phase synthesis of peptides has a long reaction cycle, large amounts of derivatized amino acids and reagents, and expensive and complex synthetic devices. Microfluidic chips, with their advantages in chemical synthesis, such as large surface area, high reaction rate, low reagent consumption, easy scale-up and integration, and high-throughput reaction, have the potential to revolutionize peptide synthesis. It is possible to achieve high-efficiency, low-consumption, low-cost peptide synthesis reactions and build high-throughput, integrated peptide synthesis systems based on them.

It is hoped that with ongoing innovation, safer and more effective peptide-mediated tumor-targeted drug delivery systems will emerge, offering new hope for patients battling refractory cancers.

## Figures and Tables

**Figure 1 pharmaceutics-16-00240-f001:**
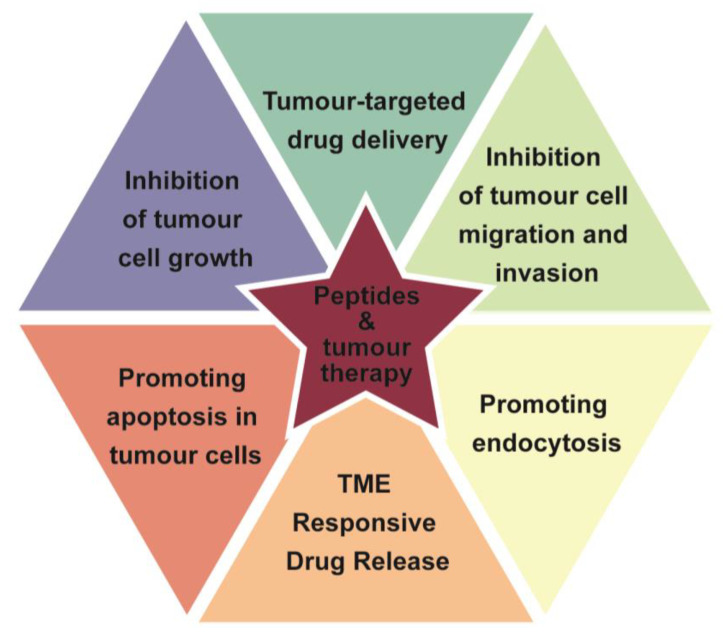
Schematic depiction of advantages of peptide-mediated drug-targeted delivery for cancer therapy.

**Figure 2 pharmaceutics-16-00240-f002:**
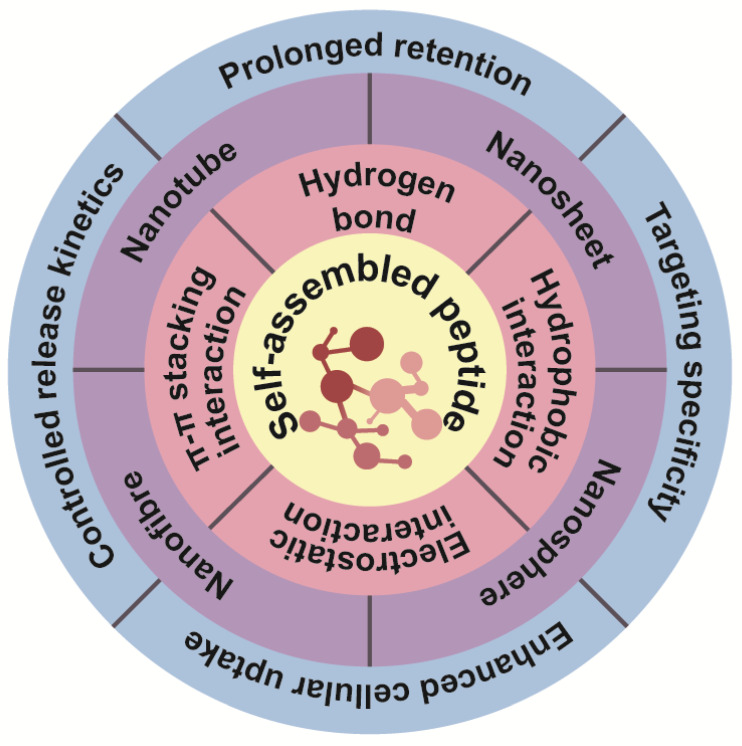
Schematic depiction of drivers, nanostructures, and advantages of self-assembled peptides.

**Figure 3 pharmaceutics-16-00240-f003:**
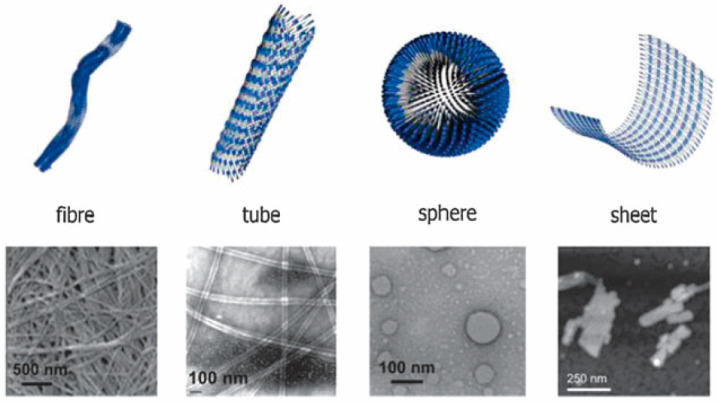
Supramolecular structures accessible via peptide self-assembly. Reprinted with permission from Ref. [53].

**Figure 5 pharmaceutics-16-00240-f005:**
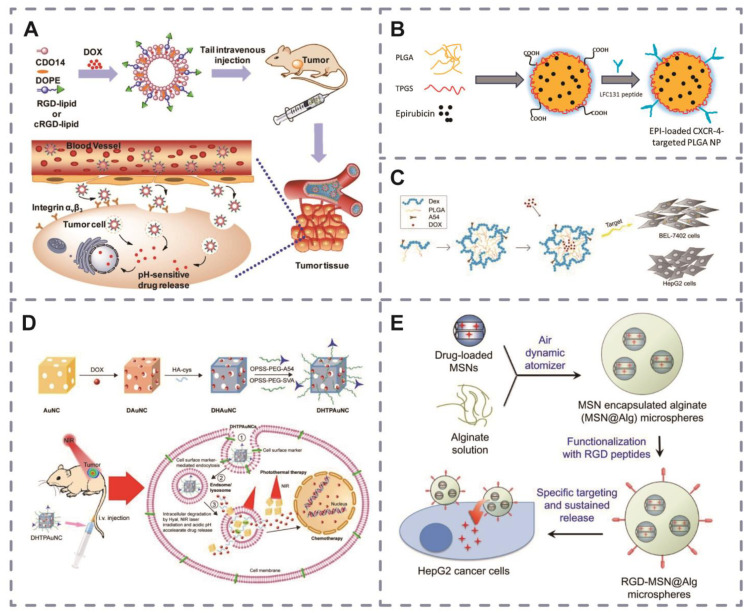
(**A**) RGD peptide-conjugated doxorubicin (DOX)-loaded liposomes for targeted cancer chemotherapy. Liposomes, including CDO14 and DOPE lipids, are modified to varying degrees through the addition of a linear RGD or cyclic cRGD conjugated lipid. The integrin α_v_β_3_ targeting liposome acquires cytotoxicity through the adsorption of DOX at the inner hydrophobic shell. Reprinted with permission from Ref. [102]. (**B**) Schematic illustration of preparation of Epirubicin-loaded PLGA/TPGS nanoparticles. The surface of the nanoparticle was conjugated with the LFC131 peptide, which is specific to the CXCR-4 receptors. Reprinted with permission from Ref. [103]. (**C**) An A54-Dex-PLGA graft was designed and synthesized, and it was easy to form polymeric micelles in aqueous solution for the delivery of DOX. Reprinted with permission from Ref. [104]. (**D**) Schematic representation for the preparation of DHTPAuNCs and hypothetical subcellular drug release behaviors and cellular uptake pathways. Reprinted with permission from Ref. [105]. (**E**) Illustration representing the drug-adsorbed mesoporous silica nanoparticles (doxorubicin-MSN) encapsulated into RGD-containing peptide-functionalized alginate microspheres. Reprinted with permission from Ref. [106].

**Figure 6 pharmaceutics-16-00240-f006:**
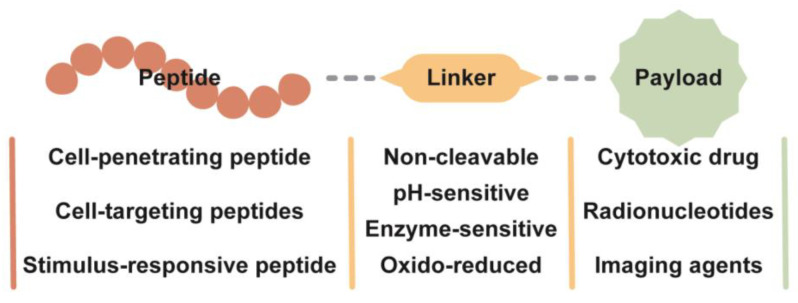
Structure of tumor-targeting drug conjugates.

**Table 1 pharmaceutics-16-00240-t001:** The role of twenty amino acids in nanomaterials.

Role in Self-Assembly	Amino Acid Name
Aliphatic hydrophobic groups provide hydrophobic forces	A, L, I, V, M
Aromatic ring hydrophobic groups provide π-π stacking	F, Y, W
Hydrophilic groups provide hydrogen bonding forces	N, Q, S, T
Charged groups provide electrostatic forces	H, R, K, E, D
Disulfide bond	C
Spatial positional resistance provides flexibility	G
Spatial positional resistance provides rigidity	P

**Table 2 pharmaceutics-16-00240-t002:** Status of PDC R&D.

Drug Name	Company	Indications	R&D Stage
Lutathera	Novartis	Gastrointestinal pancreatic	Approved
		neuroendocrine tumors	
Pepaxto	Oncopeptides	Multiple myeloma	Approved
SNG 1005	Shenogen Pharma	Brain metastatic	Phase III
	Group&Angiochem		
AN-152	AEterna Zentaris	Ovarian cancer	Phase III
EP-100	Esperance Pharmaceuticals	Breast cancer, ovarian cancer	Phase II
CBP-1008	Coherent Biopharma (CBP)	Breast cancer	Phase II
CBP-1018		Lung cancer	Phase I
BT-1718	Bicycle Therapeutics	Non-small cell lung cancer	Phase I/II
BT5528		Solid tumor	Phase I/II
BT8009		Solid tumor	Phase I/II
CBX-12	Cybrexa Therapeutics	Tumor	Phase I
TH1902	Theratechnologies	Triple-negative breast cancer	Phase I
BGC 0228	BrightGene	Advanced solid tumors	Phase I

## Data Availability

Not applicable.

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
