# Peer review of "Peptide-Mediated Nanocarriers for Targeted Drug Delivery: Developments and Strategies"

_pharmaceutics, 2024, doi:10.3390/pharmaceutics16020240_

Round 1

Reviewer 1 Report

Comments and Suggestions for Authors

The review submitted by Wang et al. is focused on the peptide-mediated drug targeted delivery. The manuscript is clear, well written and the authors have provided also some interesting perspectives. However, some minor corrections are needed:

1. as a general remark, I think that the title is misleading. From the title, I would have expected the authors to discuss only about the drug delivery systems based on peptides for tumor therapy. In reality, the drug delivery sections are quite scarce and the authors put the accent on classification and different characteristics of peptides.

2. the following references can be cited: https://doi.org/10.3390/ijms24021049

3. the authors can also discuss in the introduction section about the other types of ligands for targeted drug delivery: aptamers, antibodies, folic acid, etc. and to explain why peptides are or not superior. Some examples of ligand-functionalized carriers might be provided. 

4. line 73: revise the sentence "to a range of biological functions to obtain a series of biological functions."

5. line 103: revise: "while intermolecular bonding promotes, it promotes the.."

6. line 144: revise: "and hydrogels in aqueous solutions, and hydrogels with..."

7. line 199-202: revise this sentence

8. linen 219: revise : "peptide self-assembly, as previously discussed, peptide self-assembly..."

9. line 226: revise: "form self-assemblies of self-assemblies.."

10. line 243-247: revise the sentence

11. line 255: delete repetition "nanofibrous structures"

12. line 297-299: revise sentence

13. line 771 and 773: delete repetition.

14. line 821: revise sentence.

14. line 977: revise sentence: "encapsulation with polymeric substances offers polymeric substances that..."

Comments on the Quality of English Language

There are some repetitions which must be deleted. Also, some sentences are quite long and difficult to follow.

Reviewer 2 Report

Comments and Suggestions for Authors

The review on the subject Peptide-mediated DDS is very needed paper.

I recognize the strong effort of Authors to deliver such review.

I find it well organized and narration is very well.

However there is too many mistakes, some of them are language origin and some are simply desultory.

It took me long time to find most of them. Specifically:

1.line 52-54: .. to enhance peptide stability of the peptides… (repetition)

2.lines 85-87: The sentence has no verb

3.line 80: …form high aspect ratio nanostructures…. ? Not understandable

4.lines 113-114: …to minimize surface exposure to reduce the exposed surface area… (repetition)

5.kines 199-202: Beta-tun ….. (There are two verbs: generated…created). Please chose what you prefer

6.lines 202-203: The sentence is without verb

7.lines 216-217: Again it is repeated phrase within this sentence: Peptide self-assembly (it is twice in this sentence)

8.lines 253-256: Again repeated phrase: nanofibers structures.

9.lines 256-257: The sentence without verb, starting from Thereby. Probably you can merge these two sentences, although long sentences in such complicated text would be tiring for reader

10.line 290: Space is missing between …enzyme and [50]

11.line 303: Figure 2 or 3 ?

12.line 314: the sign ; should be rather .

13.lines 353-354: Jang et al. [71] produced thermoresponsive vesicles by exploiting the temperature responsiveness of thermoresponsive proteins (repetition again)

14.line 378-379: The diversity of amino acid species, self- 378 assembly module’s biological function modules,…… (repetition)

15.line 433: is: “without increasing toxicity”, did you mean without increasing system toxicity ?

16.lines 485-4 87: Significant differences in molecular expression that the molecular expression (Repetition again)

17. line 482: Maybe Transport instead of Transportation is more accurate ?

18.lines 527 – 530: No verb in this sentence

19.line 554-556: The sentence starting from Shi et al [108] has no verb

20.line 690: is: is mostly peptides, should be rather : are mostly peptides

21.lines 77o-774: There is repetition of the whole sentence. Remove one of them

22.line 821: There is no VERB in the sentence:

Cyclic peptides, with enhanced structural stability than straight chain peptides

23.line 839: which carboxyl group of aspartic acid: alfa or beta (add this)

24.line 935 is: In vitro experiments confirmed, the researchers concluded (no verb in first phrase)

25.line 977-979: The sentence needs rearrangement

26.line 987: even at min 4: what did you mean ?

27,line 1014: This review encapsulates; better to say includes

REGRENCES:

Journal titles are not given in: [50], [58], [73} – Revise authors list, especially he las author: E.J.o.t.N.A.o.S ?, [89], [95], [96], [104], [108], [112], [116] – what is: H.J.P.o ?, [118], [122], [125], [130], [143], [147] + V.P.J.P.o.t.N.S.o.s.?, [151] official journal ? [157], [161], [163], [165], [173], [174], [182]

Comments on the Quality of English Language

Many repeated phrases and sentnecnes without verb

Author Response

请参阅附件。

Reviewer 3 Report

Comments and Suggestions for Authors

This is a good review; I congratulate the authors on the scientific aspects of the manuscript. However, why not convert this review article from being merely good to excellent or outstanding? I would encourage the authors to consider the foregoing remarks in a positive manner; my recommendations would not take very long to undertake!

1. Please include a small section (composed of bulletin points with references) relating to the advantages and disadvantages of peptide mediated drug delivery.

2. There are some minor grammatical corrections required. The heading "hydrogen bond" should read hydrogen bonds; similarly for hydrophobic, electrostatic  etc (make them into plurals instead of singular).  

3. Line 50; instead of "This paper......" please correct to "This review...."

4.  PLEASE see the reference list below; there are very few references in this article post 2021. The reference list needs to be updated. In addition the authors are kindly invited to examine the reference list which may help them to add other useful comments in their review.

5. At the end of the review (before the "Conclusions" section) it would be useful for scientists in this and related fields to read a short paragraph,  with references,  to (current/future directions) machine learning and AI in relation to the subject matter being discussed.

6. IMPORTANT: I reiterate the comment made above. Points 1-5 would not take long to undertake/correct the current manuscript but would make it much more valuable to fellow scientists working in this field.  It is not my purpose to hold up the publication of this article.

..............................................................

Meiling Zhou The role of cell-penetrating peptides in potential anti-cancer therapy, Clin. Transl. Med., 12:e822, 2022, https://doi.org/10.1002/ctm2.822

Timothy Samec et al., Peptide-based delivery of therapeutics in cancer treatment, Mater Today Bio. 14: 100248, 2022

Aldo O. González-Cruz et al., Peptide-based drug-delivery systems: A new hope for improving cancer therapy, Journal of Drug Delivery Science and Technology, 72, 103362, 2022

Lei Wang et al., Therapeutic peptides: current applications and future directions, Signal Transduction and Targeted Therapy 7, Article number: 48 (2022)

Mingpeng Liu et al., Peptide-Enabled Targeted Delivery Systems for Therapeutic Applications

Front. Bioeng. Biotechnol., 9 - 2021 | https://doi.org/10.3389/fbioe.2021.701504

Vivek Prakash bhai Chavda et al., Peptide-Drug Conjugates: A New Hope for Cancer Management

Molecules 27(21):7232, 2022, DOI: 10.3390/molecules27217232

Ritika Sharma et al., ACS Omega 7, 41, 36092–36107, 2022, Functionalized Peptide-Based Nanoparticles for Targeted Cancer Nanotherapeutics: A State-of-the-Art Review; https://doi.org/10.1021/acsomega.2c03974, 

Sri Murugan et al., Peptides as multifunctional players in cancer therapy, Experimental & Molecular Medicine 55, 1099–1109 (2023)

.........................................................................................................

I hope that the authors find the foregoing remarks helpful rather than a hindrance!

Comments on the Quality of English Language

Only minor editing of the quality of the English language is required.

Round 2

Reviewer 2 Report

Comments and Suggestions for Authors

The reference [174] was removed. Then the reference number [174] is skipped. Please renumber the references starting from [173].

Other corrections are made satisfactorily.